# A Radio Environment Map Updating Mechanism Based on an Attention Mechanism and Siamese Neural Networks

**DOI:** 10.3390/s22186797

**Published:** 2022-09-08

**Authors:** Pan Zhen, Bangning Zhang, Chen Xie, Daoxing Guo

**Affiliations:** College of Communication Engineering, Army Engineering University of PLA, Nanjing 210001, China

**Keywords:** radio environment map, Siamese neural network, attentional mechanisms, crowdsourcing

## Abstract

A radio environment map (REM) is an effective spectrum management tool. With the increase in the number of mobile devices, the wireless environment changes more and more frequently, bringing new challenges to REM updates. Traditional update methods usually rely on the amount of data collected for updating without paying attention to whether the wireless environment has changed enough. In particular, a waste of computational resources results from the frequently updated REM when the wireless environment does not change much. When the wireless environment changes a lot, the REM is not updated promptly, resulting in a decrease in REM accuracy. To overcome the above problems, this work combines the Siamese neural network and an attention mechanism in computer vision and proposes an update mechanism based on the amount of wireless environmental change starting from image data. The method compares the newly collected crowdsourced data with the constructed REM in terms of similarity. It uses similarity to measure the necessity of the REM to be updated. The algorithm in this paper can achieve a controlled update by setting a similarity threshold with good controllability. In addition, the effectiveness of the algorithm in detecting changes of the wireless environment has been demonstrated by combing simulation data.

## 1. Introduction

A radio environment map (REM) is a full-band, full-range wireless cognitive network map containing multiple data sources, including geographic information data, radio device parameters, radio propagation models, and spectrum data, which can provide reliable and adequate knowledge of crowd sensing devices [1]. A more efficient spatial spectrum database system can provide information about the global radio. It is commonly used for various tasks in wireless communications, such as spectrum monitoring, opportunistic access, interference-free routing, and mobile network relaying [2]. REM has become an important research direction in spectrum management.

Given the increasing application scenarios of REM, in recent decades, REM with short time delay and high accuracy has become the goal. Due to the dynamics of the surrounding environment, the generated REM may become outdated, and errors gradually increase over time [3]. The dynamics of the surrounding environment can be divided into two categories, i.e., abrupt and long-term changes. Abrupt changes in the surrounding environment are mainly due to changes in the radiation sources that move in the position. Long-term changes in the surrounding environment tend to come from slow changes in temperature, buildings, vehicles, and people. Therefore, to ensure that the REM can meet the accuracy requirements in subsequent applications, updating the REM according to the current environment is necessary. In the updating process, determining that the current REM needs to be updated is the first problem to be solved. Haeberlen et al. [4] calibrated new reference signal received power (RSRP) samples at several known locations and fitted these samples to old samples in the REM to adjust the static REM. In the online phase, the estimated linear function is first used to transfer the new samples to the old samples so that the original radio diagram can be reused. Both the LANDMARC system [5] and LEASE [6] use reference points to adaptively offset changes in the RSRP samples caused by environmental changes. These systems use real-time RSRP samples received at reference points for position estimation to adapt to environmental dynamics. However, the accuracy of these methods can only be guaranteed for these systems if the reference receivers are densely distributed. In addition to this, fitting unknown point information with information from known points introduces new errors. One study [7] used the data stored in the database to determine the changes in the REM using Welch’s *t*-test. The proposed method uses the average received power difference between the RSRP data in the database at the old moment and the observed data at the new moment to determine if the REM has changed and thus decide whether it needs to be updated. In the paper [8], the authors established an access point-scoring mechanism to determine the changes in the surrounding environment by calculating the final score of the access point through user feedback on the frequency of occurrence of the access point (AP) and thus decide whether the REM needs to be updated. Both methods are used to mine the collected data to decide on the REM update.

First, the study [9] did not propose a new REM update algorithm but used robots instead of humans to perform a tedious field survey with the goal of reducing the human cost, but this requires energy and is unlikely to update the map in a short time. Secondly, some REM updating algorithms using regression algorithms have been proposed. The method in [10] uses a Gaussian process to interpolate between survey samples, thereby reducing the workload of the survey and update process. The method in [11] uses partial least-square regression (PLSR) to update radio maps using crowdsourced smartphone data. The authors reconstructed the radio map by using real-time signal strength readings received at reference points [12]. These methods thus reduce the workload of the measurement and updating process. However, they still face an important problem. The measurement updating is performed again in the offline stage because it needs to be able to identify the reference measurement location in the same way as used in the initial radio map construction. This means that the update process is still labor-intensive and needs to cover the measurements on the original reference grid points. Clearly, these methods provide the most accurate radio map updates, especially using a proper Gaussian process, but their labor requirements make them practically unusable. Finally, there are also REM update algorithms that partially use clustering algorithms. The authors of the paper [13] used a clustering algorithm for the special selection of the collected data, followed by the continuous improvement of the model using incremental learning and subsequently predicting the unknown information. They proposed the algorithm called OWUH. An adaptive wireless map algorithm based on clustering mechanism and robust regression is proposed in the paper [14]. The algorithm first builds a static radio map by collecting RSS at reference points and calibration points. Then, the path loss parameters of each reference point are calculated, and the network is clustered to divide the localization area into multiple sub-regions. Finally, robust linear regression is used to establish the relationship between RSS of reference and calibration points in each subregion.

The literature mentioned above explores the problem of REM updates. However, there are the following problems: First, current update algorithms are not based on whether the wireless environment has changed enough to make updates. They all use manual unmanned aircraft and sensing devices to collect data from the observation area to make direct updates. This way of thinking can lead to wasted resources when updates are still performed with minor changes, or significant changes in the environment are not updated in a timely manner, resulting in reduced REM accuracy. Second, using the data collected by the deployed nodes to fit the unknown region introduces new errors. If the accuracy of the fitted model is low, it will lead to high errors in the newly constructed REM. Finally, all the above studies started from the idea of using spectral data to determine the REM update.

To address the above shortcomings, this paper proposes an update mechanism for REM based on the attention mechanism and a Siamese neural network. The method borrows some ideas from computer vision and takes the perspective of the image as the starting point. A Siamese neural network (SNN) is used to determine the similarity of feature information in the already constructed REM images to the new REM constructed from the data collected by the new moment crowdsourcing method. The method is an algorithm for updating based on the detection of changes. In the subsequent use, the update threshold of the radio environment map can be set subjectively according to the radio changes in the actual environment. After experimental analysis, the algorithm proposed in this paper can realize the update by manually setting the threshold value and can effectively judge the change in the radio environment map, and its update accuracy is higher than those of other algorithms.

The rest of the article is organized as follows. Section 2 presents the knowledge and problem description of REM. Section 3 presents the algorithmic idea of the wireless environment map updating mechanism based on the SNN and the attention mechanism. Section 4 provides a relevant description of the experimental simulation. In Section 5, we provide simulation results to evaluate the performance of the proposed algorithm. Finally, the conclusions are summarized in Section 6.

## 2. System Model and Problem Description

### 2.1. Radio Environmental Map System

The REM system mainly includes crowdsourcing users, the REM system’s processing center, a spectrum database, an REM system administrator, and REM system users. First, the crowdsourcing user collects signal strength information from the surrounding area and uploads the collected data to the system’s processing center. Meanwhile, the processing center periodically sends data collection task requests to the crowdsourcing users to update the data in the database. The crowdsourcing users accept the collection tasks according to their space-time environments and transmit the system construction information to the system administrator. Next, the system administration sends the data request to the system’s processing center if requested by the system user and transmits the obtained data to the system administrator. The obtained data are transmitted to the system user. Finally, the system user provides information to the system’s manager mainly on demand to obtain valid and available REM. The system structure is shown in Figure 1. Crowdsourcing users are characterized by a large amount of data and high mobility, so in this paper, we assume that all crowdsourcing users are trusted users and will not upload false data. In addition to this, all crowdsourced user time synchronization is known.

The REM is based on the massive multidimensional spectrum data to obtain the signal strength distribution in a specific area and estimate the spectrum utilization. However, most of the existing spectrogram construction studies only describe the distribution of signal intensity in space, while ignoring frequency and time information. Ref. [15] introduced the tensor model to represent the extension work from two-dimensional spectrograms in three-dimensional or higher-dimensional spectrograms. The REM usually takes latitude and longitude as the horizontal and vertical axes. It depicts the reference signal received power at each location with different colors, as Figure 2 shows an example of the REM.

REM construction is a significant step in the REM system, and its primary purpose is to visualize regional spectrum data for easy observation, evaluation, and use. Data visualization can present the various information contained in the spectrum data more intuitively, clearly, and effectively, providing a new way of thinking about studying the spatial spectrum. In this paper, we take the perspective of using a data image as the starting point.

### 2.2. Problem Description

Spectrum information is a kind of information with high volatility in time, frequency, and space domains, which are vulnerable to the surrounding environment. Therefore, it is essential to capture the changes in the environment in time to update the existing REM. Dynamic changes in the surrounding environment can be divided into two categories, namely, abrupt and long-term changes. Abrupt changes in the surrounding environment are mainly due to changes in radiation sources that move in the position. Long-term changes in the surrounding environment tend to come from slow changes in temperature, buildings, vehicles, and people. First, we experimented with the volatility of the spectrum information in space using a real dataset provided in the paper [16]. The dataset contains actual wireless data sampled from the 2.4 GHz band. The dataset contains received signal strength indicators (RSSI) collected at five locations with a sampling interval of 100 μs, a sampling time of 5 min, and a sampling frequency of 2.401–2.483 GHz every 1 MHz. As shown in Figure 3a, we used the RSSI fluctuation plots generated at location B, with frequencies of 2.402 and 2.080 GHz and a duration of 1 ms. Figure 3b shows the graph of RSSI fluctuations generated by us using position D, with frequencies of 2.402 and 2.080 GHz and a duration of 1 ms. It can be observed from the figure that the spectrum fluctuations are different for different frequencies, different locations, and different times. Therefore, it is necessary to update the REM.

In this study, we divided the observation area into a two-dimensional grid, and the color in each grid represents the received power of the reference signal at that location. The system model is shown in Figure 4. The spectrum data were collected through crowdsourcing. The crowdsourced users first observed the radio environment, such as the receiving location and the receiving power, and uploaded the practical information to the spectrum database in real time. The spectrum database generated the initial radio environment map of the current area. The timeliness of the radio environment map decreased when the radiation source in the observed area moved, increased, or decreased; or when the surrounding environment changed. The frequency of updates varied for different areas. For example, in urban commercial areas, spectrum activities were frequent, so frequent updates were required, and for suburban environments, updates could be performed for a more extended period. To facilitate spectrum managers to make update decisions, we propose an update mechanism for REM in SNN.

## 3. A Radio Environment Map Updating Mechanism Based on the Attention Mechanism and Siamese Neural Networks

### 3.1. Realization of Ideas

First, with the spectrum database we generated an initial REM with reasonable accuracy in the observation area based on the spectrum data at time $t_{0}$. At time $t_{1}$, the system processing center was issued a data collection acquisition task. The crowdsourced users accepted the task according to their situations and then uploaded the collected data to the spectrum database. Relying on the high mobility of the crowdsourced users, we wanted to collect as much data as possible in the area, but it was challenging to collect all data from all grids in the area. Therefore, the spectrum database was filled with the received power information in the corresponding grid according to the received location uploaded by the crowdsourced users. We called the grid graph constructed from the crowdsourced data the crowdsourcing matrix. Meanwhile, the processing center segmented the original REM according to the crowdsourcing matrix constructed at moment $t_{1}$. The positions of the original REM without data in the crowdsourcing matrix were set to 0, and the positions with data in the crowdsourcing matrix were reserved. We got the segmented REM matrix and the crowdsourcing matrix. Next, we performed feature extraction on the matrix with a Siamese neural network. Finally, the similarity of the two sample matrices was output. The manager can decide whether the REM needs to be updated according to the actual situation of the observed area, combined with the similarity. The flow of the update idea is shown in Figure 5.

### 3.2. Siamese Neural Networks

SNNs [17] are coupled architectures based on two convolutional neural networks (CNNs), which take two samples as input and output their representations embedded in high-dimensional space to compare the similarity of the two samples. The SNN is formed by splicing two neural networks with the same structure and shared weights. The two neural networks perform feature extraction on the input samples by convolutional operations and subsequently map the extracted features to a uniform dimensional space to facilitate the final similarity measure. During the training of the network, the Siamese neural network maximizes the representation of different labels and minimizes the representation of the same labels under the supervised learning paradigm. Figure 6 is a schematic diagram of the Siamese neural network structure.

$X_{1}$ and $X_{2}$ denote the sample matrix of the input, $G_{W}\left(\right)$ is called the feature extraction function, and *W* denotes the hyperparameters in the network: (1)$$E_{W}\left(X_{1},X_{2}\right)=∥G_{W}\left(X_{1}\right)-G_{W}\left(X_{2}\right)∥,$$

$E_{W}\left(\right)$ is called the similarity measure. The idea of the SNN is to set the feature extraction function $G_{W}\left(\right)$ with parameter *W* and find the parameter *W* such that $E_{W}\left(X_{1},X_{2}\right)$ becomes smaller when $X_{1}$ and $X_{2}$ are in the same category and larger when $X_{1}$ and $X_{2}$ are different categories. The loss function of the network was set as
(2)$$L\left(W,{\left(Y,X_{1},X_{2}\right)}^{i}\right)=\left(1-Y\right)L_{G}\left(E_{W}{\left(X_{1},X_{2}\right)}^{i}\right)+YL_{I}\left(E_{W}{\left(X_{1},X_{2}\right)}^{i}\right),$$
where ${\left(Y,X_{1},X_{2}\right)}^{i}$ denotes the ith sample, composed of a pair of sample matrices $\left(X_{1},X_{2}\right)$ and a label *Y*. The label is usually represented using 1 or 0. If the sample matrix $\left(X_{1},X_{2}\right)$ is of the same category, the label is 1, and vice versa. $L_{G}$ is a partial loss function of the same category, and $L_{I}$ is a non-identical partial loss function.

Using SNN as the framework, we first designed two structurally identical 4-layer convolutional neural networks for feature extraction and then tiled the extracted features to the fully linked layers. The structure of the convolutional neural network is shown in Table 1.

After the neural network extracts the features, we get the feature vectors of the two input samples (Figure 7). Subsequently, the Manhattan distance (L1 distance) is calculated for the feature vectors. Finally, the similarity of the two feature vectors is obtained using the sigmoid function.

### 3.3. Attention Mechanisms

The approach in this paper is to fully extract the feature information from the already constructed REM by taking the image’s perspective as the starting point. In this method, we divide the observation area into a two-dimensional grid. Changes in the surrounding environment or radiation sources lead to changes in the observed values of part of the grid. The texture of the original REM will be corrupted. Moreover, the method proposed in this paper mainly introduces the idea of computer vision to update the REM by observing the changes in textures in the REM. Computer vision uses machines such as computers to process the input images or statements to help the human perceive more information from the images, statements, or multidimensional data to help the human make decisions. Attention mechanisms [18] have been proposed to enable computers to capture essential information in images more effectively. The attention mechanism in neural networks allocates the primary computational resources to the more critical parts. Usually, the deeper the network model, the more expressive it is; at the same time, the more parameters the model has, the more information the model stores. By introducing the attention mechanism, it is possible to focus on the more critical information in the current task and reduce the attention to other information in the input information to save computational resources and improve the efficiency and accuracy of task processing. In this study, the data collection method based on crowdsourced data cannot obtain spectrum information for all locations within the observation region, so we introduced an attention mechanism. The attention mechanism helped the neural network to be able to focus its computational attention on the data available regions and reduced the interference of data-free regions on the network. The attention mechanism is similar to human vision. It is widely used in computer vision by scanning the input information globally to obtain the target region that needs attention and subsequently devoting more resources to this region. In this paper, we used the convolutional block attention module (CBAM) [19].

The structure of CBAM is shown in Figure 8. CBAM contains two submodules: the channel attention module and the spatial attention module. We assume that the input features are
(3)$$F\in R^{C\times H\times W}.$$

The channel attention can be expressed as follows.
(4)$$M_{c}\in R^{C\times 1\times 1}.$$

Spatial attention is expressed as
(5)$$M_{s}\in R^{1\times H\times W}.$$

The refined feature is that
(6)$$\begin{array}{c} F^{\prime }=M_{c}\left(F\right)⊗F,\\ F^{\prime \prime }=M_{s}\left(F^{\prime }\right)⊗F^{{}^{\prime }}. \end{array}$$
where $F^{{}^{\prime }}$ denotes the channel features of the input features and $F^{{}^{\prime \prime }}$ denotes the final refined features. ⊗ denotes the element-by-element multiplication of the matrix.

The channel attention module first performs AvgPooling and MaxPooling on the input features to aggregate the spatial information of the input features, and we can get the AvgPooling features $F_{Avg}^{c}$ and MaxPooling features $F_{Max}^{c}$. Subsequently, the two features are passed through the multilayer perceptron (MLP). Finally, the two features output from the perceptron are summed element-by-element to obtain the channel attention feature vector. The channel attention formula can be expressed as
(7)$$M_{c}\left(F\right)=Sigmoid\left(MLP\left(AvgPool\left(F\right)\right)+MLP\left(MaxPool\left(F\right)\right)\right).$$

The spatial attention module is similar to the idea of the channel attention module in that the AvgPooling features $F_{Avg}^{s}$ and MaxPooling features $F_{Max}^{s}$ are obtained firstly. Then, the spatial attention feature vector is generated using the convolution calculation. The spatial attention formula is expressed as
(8)$$M_{s}\left(F\right)=Sigmoid\left(f^{3\times 3}\left[AvgPool\left(F\right);MaxPool\left(F\right)\right]\right),$$
where $f^{3\times 3}$ denotes the convolution kernel for the 3 × 3 convolution operation.

## 4. Experiment-Related Instructions

To evaluate the effectiveness of the proposed method, we simulated the REM update. First, we illustrate the settings of the simulation parameters. Then, we describe the simulation results.

### 4.1. Data Source

The data in this paper were obtained through MATLAB 2020b simulation. The wireless propagation model was set to draw from the paper [20] and is defined as
(9)$$P_{Rx}\left(d\right)=P_{Tx}-L\left(d_{0}\right)-10\gamma log_{10}\left(\frac{d}{d_{0}}\right)+W+F,$$
where $P_{Tx}$ is the transmit power of the radiating source, $P_{Rx}$ is the received signal power, *d* is the distance between the radiating source and the crowdsourced user, $d_{0}$ is the reference distance, γ is the path loss index, *W* denotes the log-normal shadow of the surrounding environment variation, and *F* denotes the small-scale fading. In addition, *L* is the free space loss, denoted as
(10)$$L\left(d_{0}\right)=10log_{10}{\left(\frac{4\pi d_{0}}{\lambda }\right)}^{2}.$$
λ is the wavelength, and in a natural radio environment, the shadowing deviation will be spatially correlated depending on the surrounding environment [7]. When two receiving devices receive signals emitted from the same radiation source, we modeled the shadowing correlation $\rho_{ij}$ as an exponential fading type model [21] that
(11)$$\rho_{ij}=exp\left(-\frac{△d_{ij}}{d_{cor}}ln2\right).$$
$△d_{ij}$ is the distance between two crowdsourced users *i* and *j*, and $d_{cor}$ is the correlation distance when $\rho_{ij}$ is equal to 0.5. In the paper [21], the authors obtained experimentally that $d_{cor}$ is equal to 20. Finally, the probability distribution of spatially correlated shadows was modeled as a multivariate normal distribution [22].
(12)$$f_{s}=\frac{1}{\sqrt{{\left(2\pi \right)}^{k}\Sigma }}exp\left(-\frac{1}{2}\left(x-m\right)\Sigma^{-1}\left(x-m\right)\right).$$
x denotes the average received power vector, m denotes the median path loss vector, *k* denotes the number of grids, and Σ is the shaded covariance matrix, calculated as follows: (13)$$\Sigma =\left(\begin{array}{ccc} \sigma_{1s}^{2} & \cdots & \rho_{1k}\sigma_{1s}\sigma_{ks}\\ \vdots & \ddots & \vdots \\ \rho_{k1}\sigma_{ks}\sigma_{1s} & \cdots & \sigma_{ks}^{2} \end{array}\right).$$
$\sigma_{is}$ is the standard deviation of the i-th node shading. The small-scale fading is modeled as Rayleigh fading. The simulation parameters are shown in Table 2, and the parameters were obtained from the paper [23] by testing with real data.

Finally, the total data volume was 50,000, which was divided into the training set, validation set, and test set according to the ratio of 3:1:1. Each set contained the REM of the old moment, the crowdsourcing matrix of the radiation source movement in the new moment, and the crowdsourcing matrix of the environment change in the new moment.

### 4.2. Update Method

Since this paper proposes an update mechanism for the REM, we implemented the REM update using an update method based on the forgetting factor. In this method, the received power at the new moment can be expressed as
(14)$$P_{new}=\eta P_{old}+\left(1-\eta \right)P_{collect},$$
where $P_{new}$ denotes the received power updated at the moment T; $P_{old}$ denotes the received power at moment T − 1, which is the displayed power of the current REM; and $P_{collect}$ is the received power collected and uploaded by the crowdsourcing device at the moment T. η is the set forgetting factor, and in this paper, we set it to 0.1.

### 4.3. Performance Indicators

In this paper, we use the root mean-squared error (RMSE) as a metric parameter to measure performance. The formula for calculating the mean-squared error can be expressed as
(15)$$RMSE=\sqrt{\frac{1}{N}\sum_{i=1}^{N}(V_{update}\left(i\right)-V_{true}\left(i\right))},$$
where *N* denotes the number of cells, *i* denotes the data on each cell, and $V_{update}$ and $V_{true}$ denote the updated REM and the correct REM after the change, respectively.

## 5. Experimental Analysis

To verify the performance of the algorithm based on the attention mechanism and SNN for the REM updating mechanism proposed in this paper, we compare its performance with the oblivion factor algorithm, unique averaging, and Welch’s *t*-test algorithm in the literature [7]. The horizontal and vertical coordinates of the graph are x and y, respectively, and the horizontal and vertical coordinates indicate the length and width of the monitored area.

First, we show the updated results of the REM with the example of the radiation source movement. The REM before the radiation source moves is shown in Figure 9a. We assume that the radiation source moves 200 m along the horizontal coordinate direction, and the actual REM after the movement is shown in Figure 9b. We randomly selected sample points in the changing REM to simulate the data collected by crowdsourcing and subsequently updated the REM using four methods. As shown in Figure 9c, the proposed method was used to determine whether the REM was changed and then updated. We can observe from the figure that it is similar to the actual REM after the change, without significant differences. Figure 9d shows the updating of the REM using Welch’s *t*-test algorithm. The Welch’s *t*-test was used to test whether there was a difference between the samples and thus decide whether to update. According to the observation in Figure 9d, we found that the Welch’s *t*-test method can detect the change in the REM and update the data in most of the grids. However, there is a significant difference between it and the actual REM after the change, which is because the method misjudged some grids in the change detection of the REM. Figure 9e is a direct calculation of the received power at the new moment using Equation (Equation 14), thereby updating the REM. Figure 9f shows the new REM obtained by averaging the sample data before and after the change in the radiation source. The genetic factor method and the unique averaging method used in Figure 9e,f did not accurately update the REM because they did not judge well whether the REM had changed.

Subsequently, we compared the performances of the four update algorithms. We show the relationship between the RMSE of the updated REM and the distance traveled by the radiation source in Figure 10. It can be observed from Figure 10 that the method proposed in this paper can update the REM accurately and with higher accuracy than the existing update methods. In addition, even if the moving distance of the radiation source is small, the method proposed in this paper can accurately determine the change in the REM and thus complete the update.

From Figure 10, it can be observed that the method proposed in this paper can effectively detect the changes and make updates. As the radiation source moved, the comparative RMSE gradually became larger, but the RMSE of the method proposed in this paper was basically maintained at 1. Therefore, the accuracy of the updates of the algorithm in this paper was higher than those of other algorithms. In addition, we can observe that we can accurately update the REM even if the radiation source moves a small distance.

Finally, the algorithm proposed in this paper determines the updating of the REM by the similarity of the before and after changes, so we can decide by ourselves whether to update according to the obtained similarity or not, and at the same time, we can also update automatically by setting a threshold. Figure 11 shows the RMSE comparison of our update by setting the threshold. We judged two scenarios: the radiation source moving and the environment around the radiation source changing. The “•” line in Figure 11 represents the RMSE updated by different thresholds when the radiation source was moving and the surrounding environment was not changed. The “*” line represents the RMSE updated by different thresholds when the radiation source was not moving and the surrounding environment was changed.

According to the results in Figure 11, we can see that when the threshold value is set too low, the REM does not update effectively, and the RMSE of the REM gradually increases as the environment changes. When the threshold value changes, the RMSE of the REM changes immediately, and the REM can have an effective update, which proves that the algorithm can achieve the updating of the REM through the control of the threshold value.

## 6. Conclusions

There are few studies related to updating algorithms for REM. Traditional update algorithms are not effective at detecting updates of REM; usually, the traditional methods do not detect whether the wireless environment has changed but replace the collected data, which consumes many resources. Therefore, this paper proposes a REM updating mechanism based on an SNN and the attention mechanism. First, the SNN judges the similarity between the data collected at the new moment and the REM at the old moment. The similarity is used to decide whether to update. Secondly, in crowdsourced data collection, it is impossible to obtain all the data in the observed region, so the attention mechanism is used to help the network set the region of interest with data, which improves the network’s performance while saving computational resources. Finally, the algorithm’s performance in this paper was compared with those of the existing algorithms. The experiments proved that the timing accuracy of the algorithm proposed in this paper is higher than those of other algorithms and also verified the effectiveness of REM updates that can be performed by setting the similarity threshold.

## Figures and Tables

**Figure 1 sensors-22-06797-f001:**
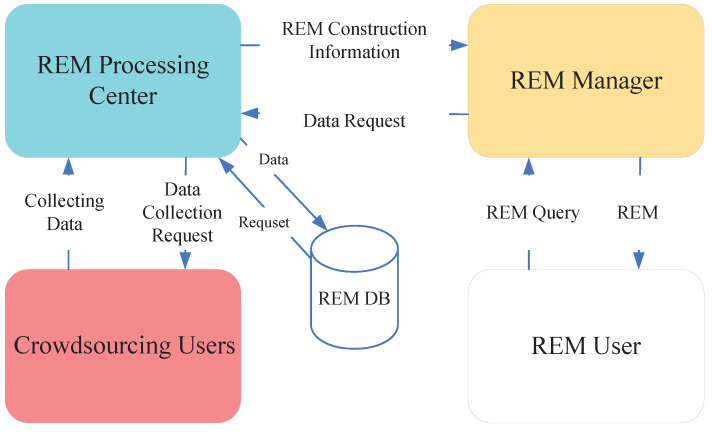
REM system structure.

**Figure 2 sensors-22-06797-f002:**
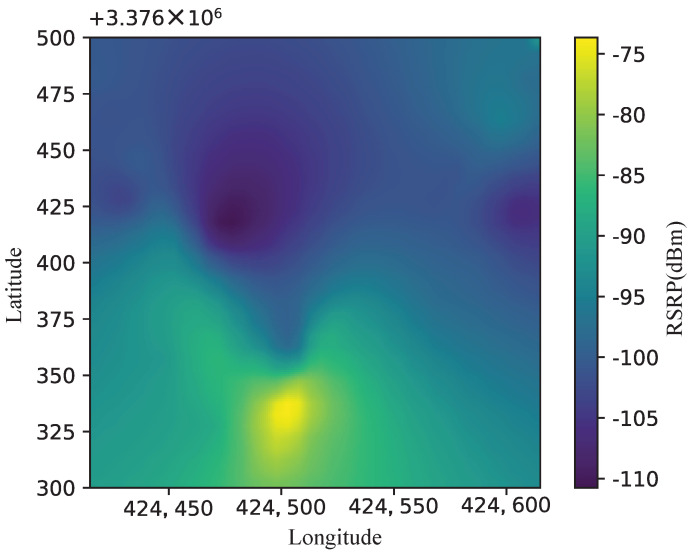
Example of REM.

**Figure 3 sensors-22-06797-f003:**
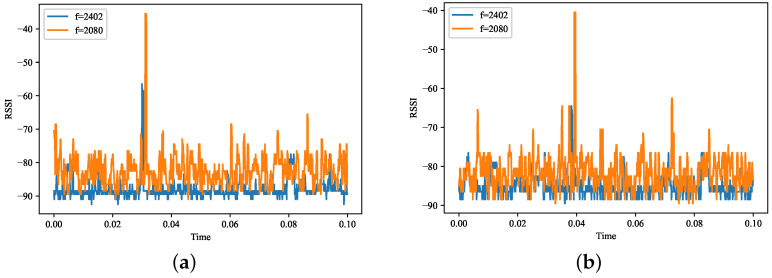
RSSI fluctuations by location and time. (**a**) RSSI chart at position B. (**b**) RSSI chart at position D.

**Figure 4 sensors-22-06797-f004:**
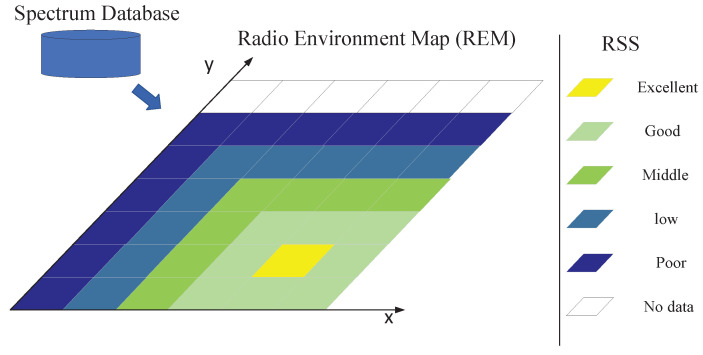
System model.

**Figure 5 sensors-22-06797-f005:**
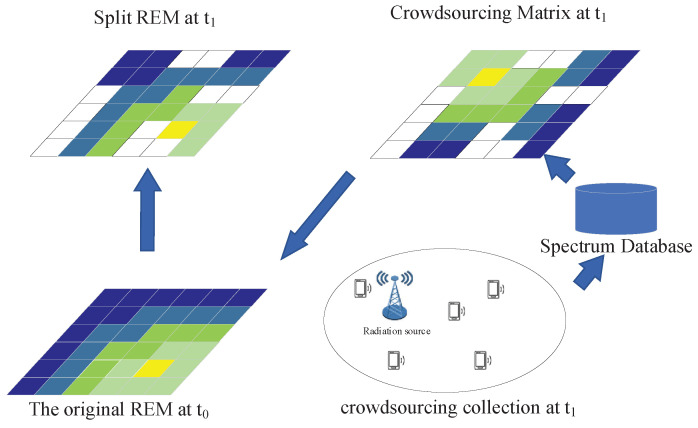
Update process.

**Figure 6 sensors-22-06797-f006:**
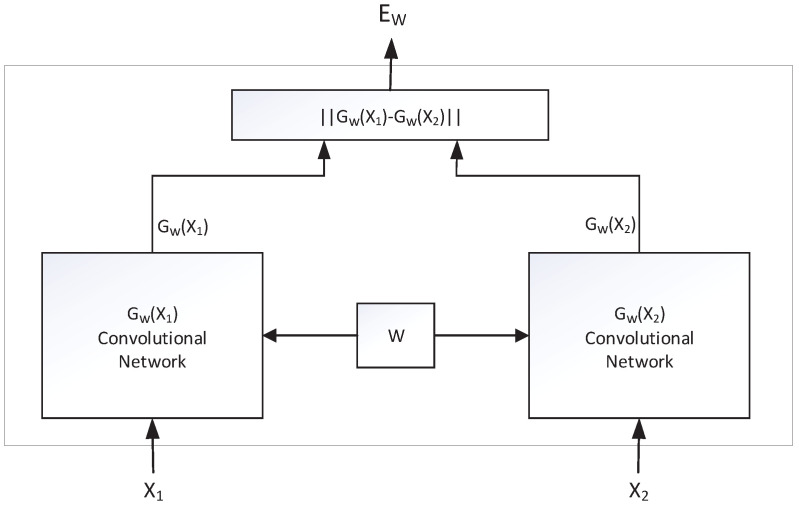
Schematic diagram of the Siamese neural network.

**Figure 7 sensors-22-06797-f007:**
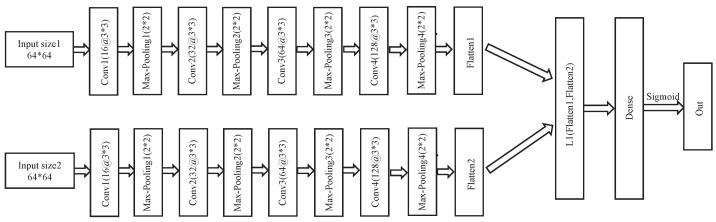
Siamese neural network model structure.

**Figure 8 sensors-22-06797-f008:**
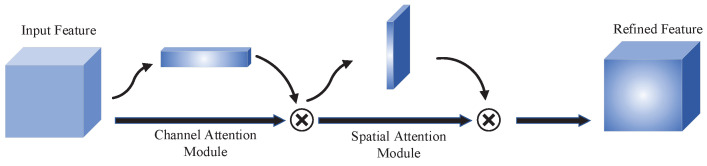
CBAM structure.

**Figure 9 sensors-22-06797-f009:**
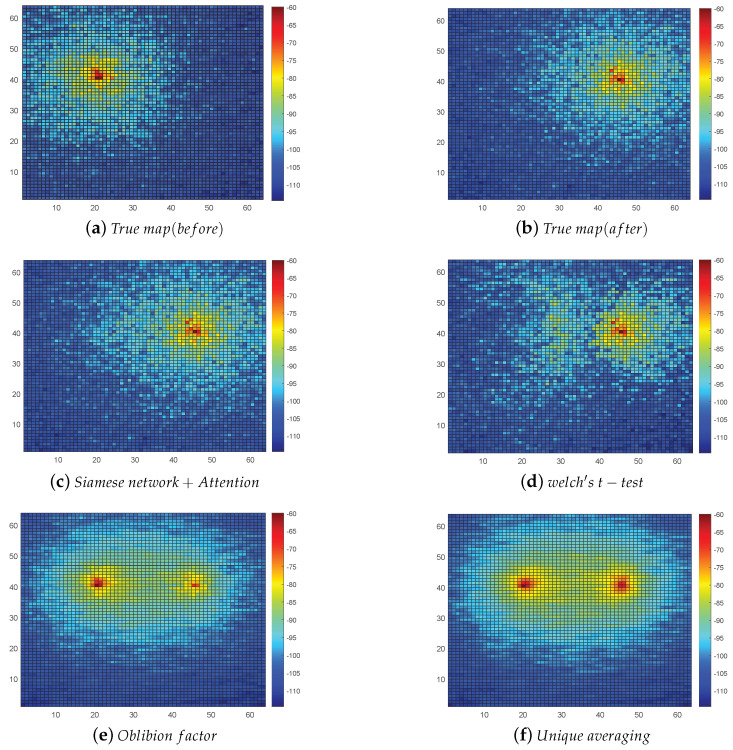
True REM and REM calculated by the four update algorithms. (**a**) Real REM before the movement of the radiation source. (**b**) Real REM after the movement of the radiation source. (**c**) REM updated by the algorithm in this paper. (**d**) REM updated by Welch’s *t*-test algorithm. (**e**) REM updated by oblivion factor algorithm. (**f**) REM updated by the unique averaging algorithm.

**Figure 10 sensors-22-06797-f010:**
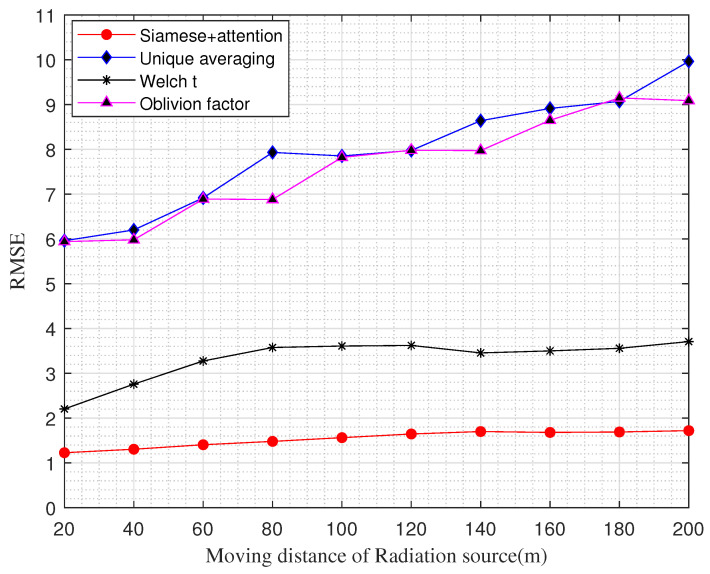
RMSE versus moving distance of the radiation source.

**Figure 11 sensors-22-06797-f011:**
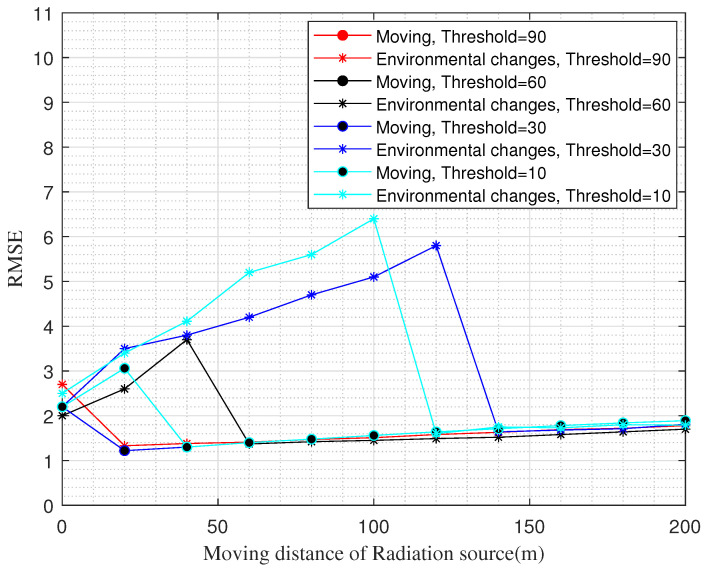
RMSE versus set threshold update.

**Table 1 sensors-22-06797-t001:** Convolutional network model details.

Input: Sample Matrix (Dimension: 64 × 64)	
Layers	Kernel Size
Conv1 + Relu	16 × 3 × 3
Max-Pooling1	2 × 2
Conv2 + Relu	32 × 3 × 3
Max-Pooling2	2 × 2
Conv3 + Relu	64 × 3 × 3
Max-Pooling3	2 × 2
Conv4 + Relu	128 × 3 × 3
Max-Pooling4	2 × 2
Flatten	-
**Output: Feature vector (Dimension: 128 × 1)**	

**Table 2 sensors-22-06797-t002:** Simulation parameters.

Parameters	Symbol	Value
Observation Area N	×N [$m^{2}$]	640 × 640
Grid size	n × n [$m^{2}$]	10 × 10
Transmitting power	$P_{Tx}$ [dBm]	29
Path loss factor	γ	4.5
Reference Distance	$d_{0}$ [m]	10
Standard shading deviation	$\sigma_{is}$ [dB]	8
Related distances	$d_{cor}$ [m]	20
Center Frequency	FB [MHz]	3500

## Data Availability

Not applicable.

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
