# Peer review of "A Radio Environment Map Updating Mechanism Based on an Attention Mechanism and Siamese Neural Networks"

_sensors, 2022, doi:10.3390/s22186797_

Round 1

Reviewer 1 Report

The manuscript proposes a new approach for REM updates based on a neural network to reduce the reconstitution error while adjusting the updates depending on the temporal evolution of the environment.

The main idea is sufficiently clear and potentially interesting. However, the overall quality is way below the bar for a scientific publication. In the current form, it is tough to read. The language is poor, and the paper is full of typos. This issue concerns both grammar and sentence structure. Professional proofreading services should be considered. In this form, it is hard for me to provide comments on the technical contribution. Below are some examples (among the many) to help the authors to understand some issues; but, again, a very careful rewriting of some parts is necessary.

The sentence "A radio environment map (REM) is a full-band, full-range wireless cognitive network map containing multiple data sources, including geographic information data, radio device coefficients, radio propagation models, and spectrum data, which can provide perfect,reliable, and adequate knowledge of intelligent device systems[1]. " is quite misleading:

  1. What are radio device coefficients?
  2. How can a map be perfect? We are talling about estimation, updates, etc. I don't think this is the right adjective.
  3. What are the "intelligent device systems"? Do they really need to be intelligent to be included in REM?

Make sure that there is a blank space between a word and a reference, i.e., "device[x]." should be "device [x]."

The sentence "samples at several known locations and fitted the linearity between these samples and the old samples in the radio map function to adjust the static"

is wrong. I understand the fitting is linear (first order) but the way it is written is not correct.

The authors should easily detect mistakes like "In the literature citeLimJS (2013)" before submission.

The meaning of the sentence: "The time geographic information data is reflected in the organic collection of massive multi-bit spectrum data" is vague. What is massive multi-bit?

The meaning of the sentence: "The REM usually takes latitude and longitude as the horizontal and vertical axes. " is unclear. The authors mix x-y axes with x-z axes.

The following sentence is unreadable "During the training of the network, the siamese neural network maximizes the representation of different labels and minimizes the representation of the same labels under the supervised learning paradigm, and minimizes the representation between the original input and the interference input under the self-supervised or unsupervised learning paradigm. "

Reviewer 2 Report

1. The authors use Siamese neural network to detect changes in radio map. This is novel if no other related works exist in the radio map context. 

1) more exhaustive literature survey is needed to confirm this.

2) Besides SNN, Are there any other methods that can be used to detect image changes? Some related references or different NN architecture should be added/discussed.

2. The interference aspects are not explicitly discussed. For example, if multiple sources exist in the considered area, what's its impact on the general problem, and your proposed approach?

3. The data comes from crowdsourcing users (CUs). 

1) The location distribution of CUs and its impact are not discussed in the main text and in the simulation setup. Fixed or moving users?

2) The time synchronization among CUs: how? Impact?

4. The applied method, CBAM, is used without sufficient justification. Why do you choose this method? Any other attension mechanism? Pros and cons? What's the unique technical essence about CBAM?

5. Is it narrowband or wideband? How do you deal with small scale fading? Simple averaging, how long? 

Some typos, cite items, figures numbers, not displaying properly.

Round 2

Reviewer 2 Report

No further comments. English writing should be improved.